# Learning Stable Deep Dynamics Models

**Gaurav Manek**
Department of Computer Science
Carnegie Mellon University
gmanek@cs.cmu.edu

**J. Zico Kolter**
Department of Computer Science
Carnegie Mellon University
and Bosch Center for AI
zkolter@cs.cmu.edu

## Abstract

Deep networks are commonly used to model dynamical systems, predicting how the state of a system will evolve over time (either autonomously or in response to control inputs). Despite the predictive power of these systems, it has been difficult to make formal claims about the basic properties of the learned systems. In this paper, we propose an approach for learning dynamical systems that are guaranteed to be stable over the entire state space. The approach works by jointly learning a dynamics model and Lyapunov function that guarantees non-expansiveness of the dynamics under the learned Lyapunov function. We show that such learning systems are able to model simple dynamical systems and can be combined with additional deep generative models to learn complex dynamics, such as video textures, in a fully end-to-end fashion.

## 1 Introduction

This paper deals with the task of learning (continuous time) dynamical systems. That is, given a state at time $t$, $x(t) \in \mathbb{R}^n$ we want to model the time-derivative of the state

$$\dot{x}(t) \equiv \frac{d}{dt}x(t) = f(x(t)) \tag{1}$$

for some function $f : \mathbb{R}^n \to \mathbb{R}^n$. Modeling the time evolution of such dynamical systems (or their counterparts with control inputs $\dot{x}(t) = f(x(t), u(t))$ for $u(t) \in \mathbb{R}^m$) is a foundational problem, with applications in reinforcement learning, control, forecasting, and many other settings. Owing to their representational power, neural networks have long been a natural choice for modeling the function $f$ [7, 14, 13, 6]. However, when using a generic neural network to model dynamics in this setting, very little can be guaranteed about the behavior of the learned system. For example, it is extremely difficult to say anything about the *stability* properties of a learned model (informally, the tendency of the system to remain within some invariant bounded set). While some recent work has begun to consider stability properties of neural networks [5, 17, 19], it has typically done so by ("softly") enforcing stability as an additional loss term on the training data. Consequently, they can say little about the stability of the system in unseen states.

In this paper, we propose an approach to learning neural network dynamics that are *provably* stable over the entirety of the state space. To do so, we jointly learn the system dynamics and a Lyapunov function. This stability is a hard constraint imposed upon the model: unlike recent approaches, we do not enforce stability via an imposed loss function but build it directly into the dynamics of the model (i.e. Even a randomly initialized model in our proposed model class will be provably stable everywhere in state space). The key to this is the design of a proper Lyapunov function, based on input convex neural networks [1], which ensures global exponential stability to an equilibrium point while still allowing for expressive dynamics.

Using these methods, we demonstrate learning dynamics of physical models such as $n$-link pendulums, and show a substantial improvement over generic networks. We also show how such dynamics models can be integrated into larger network systems to learn dynamics over complex output spaces. In particular, we show how to combine the model with a variational auto-encoder (VAE) [11] to learn dynamic "video textures" [18].

## 2 Background and related work

**Stability of dynamical systems.** Our work primarily considers the setting of autonomous dynamics systems $\dot{x}(t) = f(x(t))$ for $x(t) \in \mathbb{R}^n$. (The methods are applicable to the dynamics with control as well, but we focus on the autonomous case for simplicity of exposition.) Such a system is defined to be *globally asymptotically stable* (for simplicity, around the equilibrium point $x_e = 0$) if we have $x(t) \to 0$ as $t \to \infty$ for any initial state $x(0) \in \mathbb{R}^n$; $f$ is *locally asymptotically stable* if the same holds but only for $x(0) \in B$ where $B$ is some bounded set containing the origin. Similarly, $f$ is *globally (locally, respectively) exponentially stable* (i.e., converges to the equilibrium "exponentially quickly") if

$$\|x(t)\|_2 \le m\|x(0)\|_2 e^{-\alpha t} \tag{2}$$

for some constants $m, \alpha \ge 0$ for any $x(0) \in \mathbb{R}^n$ ($B$, respectively).

The area of Lyapunov theory [9, 12] establishes the connection between the various types of stability mentioned above and descent according to a particular type of function known as a Lyapunov function. Specifically, let $V : \mathbb{R}^n \to \mathbb{R}$ be a continuously differentiable positive definite function, i.e., $V(x) > 0$ for $x \neq 0$ and $V(0) = 0$. Lyapunov analysis says that $f$ is stable (according to the different definitions above), if and only if we can find some function $V$ as above such the *value of this function is decreasing along trajectories generated by $f$*. Formally, this is the condition that the time derivative $\dot{V}(x(t)) < 0$, i.e.,

$$\dot{V}(x(t)) \equiv \frac{d}{dt}V(x(t)) = \nabla V(x)^T \frac{d}{dt}x(t) = \nabla V(x)^T f(x(t)) < 0 \tag{3}$$

This condition must hold for all $x(t) \in \mathbb{R}^n$ or for all $x(t) \in B$ to ensure global or local stability respectively. Similarly $f$ is globally asymptotically stable if and only if there exists positive definite $V$ such that

$$\dot{V}(x(t)) \le -\alpha V(x(t)), \quad \text{with } c_1\|x\|_2^2 \le V(x) \le c_2\|x\|_2^2. \tag{4}$$

Showing that these conditions imply the various forms of stability is relatively straightforward, but showing the converse (that any stable system must obey this property for some $V$) is relatively more complex. In this paper, however, we are largely concerned with the "simpler" of these two directions, as our goal is to enforce conditions that ensure stability.

**Stability of linear systems.** For a linear system with matrix $A$:

$$\dot{x}(t) = Ax(t) \tag{5}$$

it is well-established that the system is stable if and only if the real components of the the eigenvalues of $A$ are all strictly negative ($\mathsf{Re}(\lambda_i(A)) < 0$). Equivalently, the same same property can be shown via a positive definite quadratic Lyapunov function

$$V(x) = x^T Q x \tag{6}$$

for $Q \succ 0$. In this case, by Equation 4, the following ensures stability:

$$\dot{V}(x(t)) = x(t)^T A^T Q x(t) + x(t)^T Q A x(t) \le -\alpha x(t)^T Q x(t) \tag{7}$$

i.e., if we can find a positive definite matrix $Q \succeq I$ with that $A^T Q + QA + \alpha Q \preceq 0$ negative semidefinite. Such bounds (and much more complex extensions) for the basis for using linear matrix inequalities (LMIs), as a method to ensure stability of linear dynamical systems. The methods also have applicability to non-linear systems, and several authors have used LMI analysis to learn non-linear dynamical systems by constraining the linearization of the systems to have global Lyapunov functions [10, 2, 20],

The point we want to emphasize from the above discussion, though, is that the task of *learning* even a stable *linear* dynamical system is not a convex problem. Although the constraints

$$Q \succeq I, \quad A^T Q + QA + \alpha Q \preceq 0 \tag{8}$$

are convex in $A$ and $Q$ separately, they are not convex in $A$ and $Q$ jointly. Thus, the problem of jointly learning a stable linear dynamical system and its corresponding Lyapunov function, even for the simple linear-quadratic setting, is *not* a convex optimization problem, and alternative techniques such as alternating minimization need to be employed instead. Alternatively, past work has also looked at different heuristics, such as approximately projecting a dynamics function $A$ onto the (non-convex) stable set of matrices with eigenvalues $\mathrm{Re}(\lambda_i(A)) < 0$ [3].

**Stability of non-linear systems**    For general non-linear systems, establishing stability via Lyapunov techniques is typically even more challenging. For the typical task here, which is that of establishing stability of some *known* dynamics $\dot{x}(t) = f(x(t))$, finding a suitable Lyapunov function is often more an art than a science. Although some general techniques such as sum-of-squares certification [16, 15] provide general methods for certifying stability of e.g., polynomial systems, these are often expensive and don't easily scale to high dimensional systems.

Notably, our proposed approach here is able to learn provably stable systems without solving this (generally hard) problem. Specifically, while it is difficult to find a Lyapunov function that certifies the stability of some *known* system, we exploit the fact that it is relatively much easier to *enforce* some function to behave in a stable manner according to a Lyapunov function.

**Lyapunov functions in deep learning**    Finally, there has been a small set of recent work exploring the intersection of deep learning and Lyapunov analysis [5, 17, 19]. Although related to our work here, the approach in this past work is quite different. As is more common in the control setting, these papers try to learn neural-network-based Lyapunov functions for control policies, but in way that enforces stability via a loss penalty. For instance Richards et al., [17] optimize a loss function that encourages $\dot{V}(x) \leq 0$ for $x$ in some training set. In contrast, our work guarantees absolute stability *everywhere* in the state space, not just at a small set of points; but only for a simpler setting where the *entire* dynamics are to be learned (and hence can be "forced" to be stable) rather than a stabilizing controller for known dynamics.

## 3   Joint learning of dynamics and Lyapunov functions

The intuition of the approach we propose in this paper is straightforward: instead of learning a dynamics function and attempting to separately verify its stability via a Lyapunov function, we propose to *jointly learn a dynamics model and Lyapunov function, where the dynamics is inherently constrained to be stable (everywhere in the state space) according to the Lyapunov function.*

Specifically, following the principles mentioned above, let $\hat{f} : \mathbb{R}^n \to \mathbb{R}^n$ denote a "nominal" dynamics model, and let $V : \mathbb{R}^n \to \mathbb{R}$ be a positive definite function: $V(x) \geq 0$ for $x \neq 0$ and $V(0) = 0$. Then in order to (provably, globally) ensure that a dynamics function is stable, we can simply project $\hat{f}$ such that it satisfies the condition

$$\nabla V(x)^T \hat{f}(x) \leq -\alpha V(x) \tag{9}$$

i.e., we define the dynamics

$$
\begin{aligned}
f(x) &= \mathsf{Proj}\left( \hat{f}(x), \{f : \nabla V(x)^T f \leq -\alpha V(x)\} \right) \\
&= \begin{cases} \hat{f}(x) & \text{if } \nabla V(x)^T \hat{f}(x) \leq -\alpha V(x) \\ \hat{f}(x) - \nabla V(x) \frac{\nabla V(x)^T \hat{f}(x) + \alpha V(x)}{\|\nabla V(x)\|_2^2} & \text{otherwise} \end{cases} \\
&= \hat{f}(x) - \nabla V(x) \frac{\mathsf{ReLU}\left( \nabla V(x)^T \hat{f}(x) + \alpha V(x) \right)}{\|\nabla V(x)\|_2^2}
\end{aligned}
\tag{10}
$$

where $\mathsf{Proj}(\mathsf{x}; \mathcal{C})$ denotes the orthogonal projection of $x$ onto the point $\mathcal{C}$, and where the second equation follows from the analytical projection of a point onto a halfspace. As long as $V$ is defined using automatic differentiation tools, it is straightforward to include the gradient $\nabla V$ terms into the definition of $f$, and our final network can be trained just like any other function. The general approach here is illustrated in Figure 1.

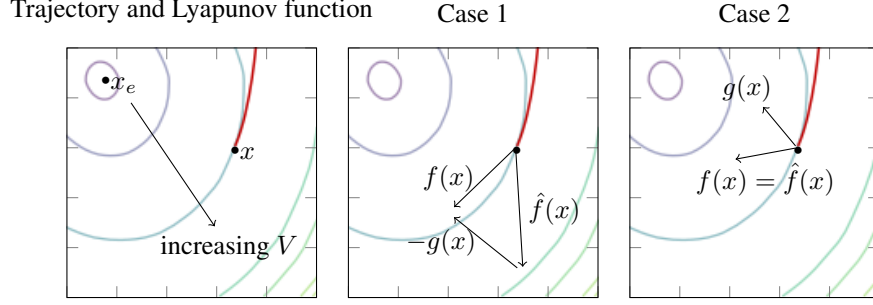

Figure 1: We plot the trajectory and the contour of a Lyapunov function of a stable dynamical system and illustrate our method. Let $g(x) = \frac{\nabla V(x)}{\|\nabla V(x)\|_2^2} \text{ReLU}\left(\nabla V(x)^T \hat{f}(x) + \alpha V(x)\right)$. In the first case $\hat{f}(x)$ has a component $g(x)$ not in the halfspace, which we subtract to obtain $f(x)$. In the second case $\hat{f}(x)$ is already in the halfspace, so is returned unchanged.

### 3.1    Properties of the Lyapunov function $V$

Although the treatment above seems to make the problem of learning stable systems quite straight-forward, the sublety of the approach lies in the choice of the function $V$. Specifically, as mentioned previously, $V$ needs to be positive definite, but additionally $V$ needs to have *no local optima except* 0. This is due to Lyapunov decrease condition: recall that we are attempting to guarantee stability to the equilibrium point $x = 0$, yet the decrease condition imposed upon the dynamics means that $V$ is decreasing along trajectories of $f$. If $V$ has a local optimum away from the origin, the dynamics can in theory get stuck in this location; this manifests itself by the $\|\nabla V(x)\|_2^2$ term going to zero, which results in the dynamics becoming undefined at the optima.

To enforce these conditions, we make the following design decisions regarding $V$:

**No local optima.**    We represent $V$ via an input-convex neural network (ICNN) function $g$ [1], which enforces the condition that $g(x)$ be convex in its inputs $x$. A fairly generic form of such networks consists is given by the recurrence

$$
\begin{aligned}
z_1 &= \sigma_0(W_0 x + b_0) \\
z_{i+1} &= \sigma_i(U_i z_i + W_i x + b_i), i = 1, \ldots, k-1 \\
g(x) &\equiv z_k
\end{aligned}
\tag{11}
$$

where $W_i$ are real-valued weights mapping from inputs to the $i + 1$ layer activations; $U_i$ are *positive* weights mapping previously layer activations $z_i$ to the next layer; $b_i$ are real-valued biases; and $\sigma_i$ are *convex, monotonically non-decreasing* non-linear activations, such as the ReLU or smooth variants. It is straightforward to show that with this formulation, $g$ is convex in $x$ [1], and indeed any convex function can be approximated by such networks [4].

**Positive definite.**    While the ICNN property can enforce that $V$ have only a single global optima, it does not necessarily enforce that this optima be at $x = 0$. While one could fix this by e.g., removing the biases term (but this imposes substantial limitations on the representable functions, which can no longer be arbitrary convex functions) or by shifting whatever global minima exists to the origin (but this requires finding the global minimum during training, which itself is computationally expensive), we take an alternative approach and simply shift the function such that $V(0) = 0$, and add a small quadratic regularization term to ensure strict positive definiteness.

$$
V(x) = \sigma_{k+1}(g(x) - g(0)) + \epsilon \|x\|_2^2.
\tag{12}
$$

where $\sigma_k$ is a *positive* convex non-decreasing function with $\sigma_k(0) = 0$, $g$ is the ICNN defined previously, and $\epsilon$ is a small constant. These terms together still enforce (strong) convexity and positive definiteness of $V$.

**Continuously differentiable.**    Although not always required, several of the conditions for Lyapunov stability are simplified is $V$ is continuously differentiable. To achieve this, rather than use ReLU

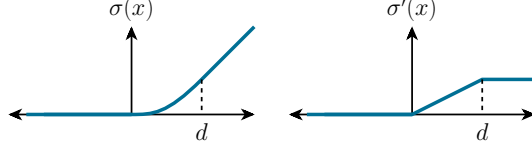

Figure 2: Smoothed ReLU, used to make our Lyapunov function continuously differentiable.

activations,[1] we use a smoothed version that replaces the purely linear ReLU with a quadratic region in $[0, d]$

$$\sigma(x) = \begin{cases} 0 & \text{if } x \leq 0 \\ x^2/2d & \text{if } 0 < x < d \\ x - d/2 & \text{otherwise} \end{cases} \quad (13)$$

An illustration of this activation is shown in Figure 2.

**(Optional) Warped input space.** Although convexity ensures that the Lyapunov function have no local optima, this is a sufficient but not necessary condition, and indeed requiring a strongly convex Lyapunov function may impose too strict a requirement upon the learned dynamics. For this reason, the input to the ICNN function $g(x)$ above can be optionally preceded by any continuously differentiable *invertible* function $F : \mathbb{R}^n \times \mathbb{R}^n$, i.e., using

$$V(x) = \sigma_{k+1}(g(F(x)) - g(F(0))) + \epsilon \|x\|_2^2. \quad (14)$$

as the Lyapunov function. Invertibility ensures that the sublevel sets of $V$ (which are convex sets, by definition) map to contiguous regions of the composite function $g \circ F$, thus ensuring that no local optima exist in this composed function.

With these conditions in place, we have the following result.

**Theorem 1.** *The dynamics defined by*

$$\dot{x} = f(x) \quad (15)$$

*defined by $f$ from (10) and $V$ from (12) or (14) are globally exponentially stable to the equilibrium point $x = 0$, for any (bounded weight) networks defining the $\hat{f}$ and $V$ functions.*

*Proof.* The proof is straightforward, and relies on the properties of the networks created above. First, note that by our definitions we have, for some $M$,

$$\epsilon \|x\|_2^2 \leq V(x) \leq M \|x\|_2^2 \quad (16)$$

where the lower bound follows by definition and the fact that $g$ is positive. The upper bound follows from the fact that the $\sigma$ activation as defined is linear for large $x$ and quadratic around 0. This fact in turn implies that $V(x)$ behaves linearly as $\|x\| \to \infty$, and is quadratic around the origin, so can be upper bounded by some quadratic $M \|x\|_2^2$.

The fact the $V$ is continuously differentiable means that $\nabla V(x)$ (in $f$) is defined everywhere, bounds on $\|\nabla V(x)\|_2^2$ for all $x$ follows from the the Lipschitz property of $V$, the fact that $0 \leq \sigma'(x) \leq 1$, and the $\epsilon \|x\|_2^2$ term

$$\epsilon \|x\|_2 \leq \|\nabla V(x)\|_2 \leq \sum_{i=1}^{k} \prod_{j=i}^{k} \|U_j\|_2 \|W_i\|_2 \quad (17)$$

where $\|\cdot\|_2$ denotes the operator norm when applied to a matrix. This implies that the dynamics are defined and bounded everywhere owing to the choice of function $\hat{f}$.

Now, consider some initial state $x(0)$. The definition of $f$ implies that

$$\frac{d}{dt} V(x(t)) = \nabla V(x)^T \frac{d}{dt} x(t) = \nabla V(x)^T f(x) \leq -\alpha V(x(t)). \quad (18)$$

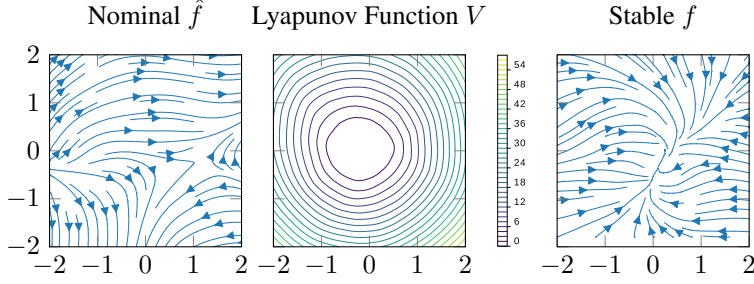

Figure 3: (left) Nominal dynamics $\hat{f}$ for random network; (center) Convex positive definite Lyapunov function generated by random ICNN with constraints from Section 3.1; (right) Resulting stable dynamics $f$.

Integrating this equation gives the bound

$$V(x(t)) \leq V(x(0))e^{-\alpha t} \tag{19}$$

and applying the lower and upper bounds gives

$$\epsilon\|x(t)\|_2^2 \leq M\|x(0)\|_2^2 e^{-\alpha t} \implies \|x(t)\|_2 \leq \frac{M}{\epsilon}\|x(0)\|_2 e^{-\alpha t/2} \tag{20}$$

as required for global exponential convergence. $\qquad\square$

## 4 Empirical results

We illustrate our technique on several example problems, first highlighting the (inherent) stability of the method for random networks, demonstrating learning on simple $n$-link pendulum dynamics, and finally learning high-dimensional stable latent space dynamics for dynamic video textures via a VAE model.

### 4.1 Random networks

Although we mention this only briefly, it is interesting to visualize the dynamics created by random networks according to our process, i.e., before any training at all. Because the dynamics models are inherently stable, these random networks lead to stable dynamics with interesting behaviors, illusrated in Figure 3. Specifically, we let $\hat{f}$ be defined by a 2-100-100-2 fully connected network, and $V$ be a 2-100-100-1 ICNN, with both networks initialized via the default weights of PyTorch (the Kaiming uniform initialization [8]) and with the ICNN having it's $U$ weights further put through a softplus unit to make them positive.

### 4.2 $n$-link pendulum

Next we look at the ability of our approach to model a physically-based dynamical system, specifically the $n$-link pendulum. A damped, rigid $n$-link pendulum's state $x$ can be described by the angular position $\theta_i$ and angular velocity $\theta_i$ of each link $i$. As before $\hat{f}$ is a $2n$-100-100-$2n$ network, and the Lyapunov function $V$ is a $2n$-60-60-1 ICNN with properties described in Section 3.1. Models are trained with pairs of data $(x, \dot{x})$ produced by the symbolic algebra solver sympy, using simulation code adapted from [21].

In Figure 4, we compare the simulated dynamics with the learned dynamics in the case of a simple damped pendulum (i.e. with $n = 1$), showing both the streamplot of the vector field and a single simulated trajectory, and draw a contour plot of the learned Lyapunov function. As seen, the system is able to learn dynamics that can accurately predict motion of the system even over long time periods.

We also evaluate the learned dynamics quantitatively varying $n$ and the time horizon of simulation. Figure 5 presents the total error over time for the 8-link pendulum, and the average cumulative error over 1000 time steps for different values of $n$. While both the simple and our stable models show

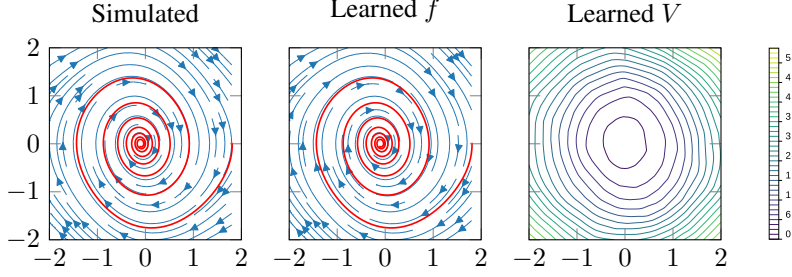

Figure 4: Dynamics of a simple damped pendulum. From left to right: the dynamics as simulated from first principles, the dynamics model $f$ learned by our method, and the Lyapunov function $V$ learned by our method (under which $f$ is non-expansive).

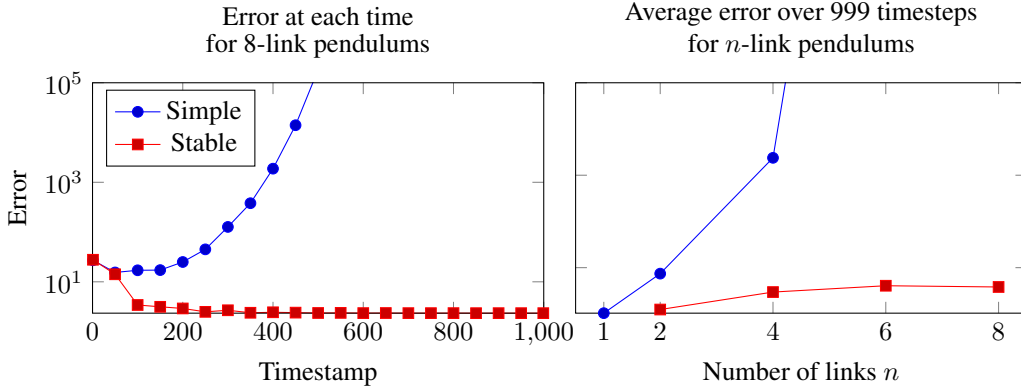

Figure 5: Error in predicting $\theta, \dot{\theta}$ in 8-link pendulum at each timestep (left); and average error over 999 timesteps as the number of links in the pendulum increases (right).

increasing mean error at the start of the trajectory, our model is able to capture the contraction in the physical system (implied by conservation of energy) and in fact exhibits decreasing error towards the end of the simulation (the true and simulated dynamics are both stable). In comparison, the error in the simple model increases.

### 4.3 Video Texture Generation

Finally, We apply our technique to stable video texture generation, using a Variational Autoencoder (VAE) [11] to learn an encoding for images, and our stable network to learn a dynamics model in encoding-space. Given a sequence of frames $(y_0, y_1, \ldots)$, we feed the network the frame at time $t$ and train it to reconstruct the frames at time $t$ and $t + 1$. Specifically, we consider a VAE defined by the encoder $e : \mathcal{Y} \to \mathbb{R}^{2n}$ giving mean and variance $\mu, \log \sigma_t^2 = e(y_t)$, latent state $z_t \in \mathbb{R}^n \sim \mathcal{N}(\mu_t, \sigma_t^2)$, and decoder $d : \mathbb{R}^n \to \mathcal{Y}, y_t \approx d(z_t)$. We train the network to minimize both the standard VAE loss (reconstruction error plus a KL divergence term), but *also* minimize the reconstruction loss of a next predicted state. We model the evolution of the latent dynamics at $z_{t+1} \approx f(z_t)$, or more precisely $y_{t+1} \approx d(f(z_t))$. In other words, as illustrated in Figure 6, we train the full system to minimize

$$\underset{e,d,\hat{f},V}{\text{minimize}} \sum_{t=1}^{T-1} \left( \mathsf{KL}(\mathcal{N}(\mu_t, \sigma_t^2 I \| \mathcal{N}(0, I)) + \mathbf{E}_z \left[ \|d(z_t) - y_t\|_2^2 + \|d(f(z_t)) - y_{t+1}\|_2^2 \right] \right) \quad (21)$$

We train the model on pairs of successive frames sampled from videos. To generate video textures, we seed the dynamics model with the encoding of a single frame and numerically integrate the dynamics model to obtain a trajectory. The VAE decoder converts each step of the trajectory into a frame. In Figure 7, we present sample stable trajectories and frames produced by our network. For comparison, we also include an example trajectory and resulting frames when the dynamics are modelled without the stability constraint (i.e. letting $f$ in the above loss be a generic neural network). For the naive

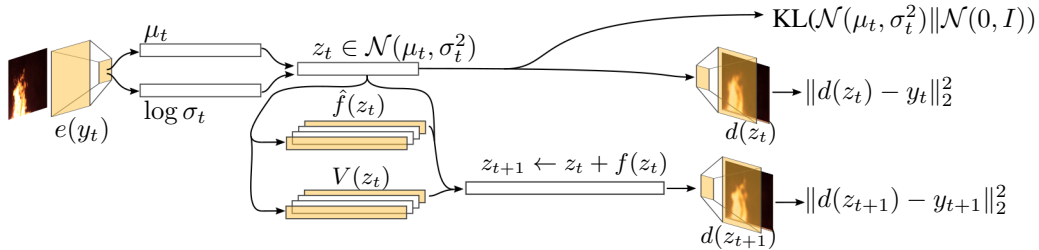

Figure 6: Structure of our video texture generation network. The encoder $e$ and decoder $d$ form a Variational Autoencoder, and the stable dynamics model $f$ is trained together with the decoder to predict the next frame in the video texture.

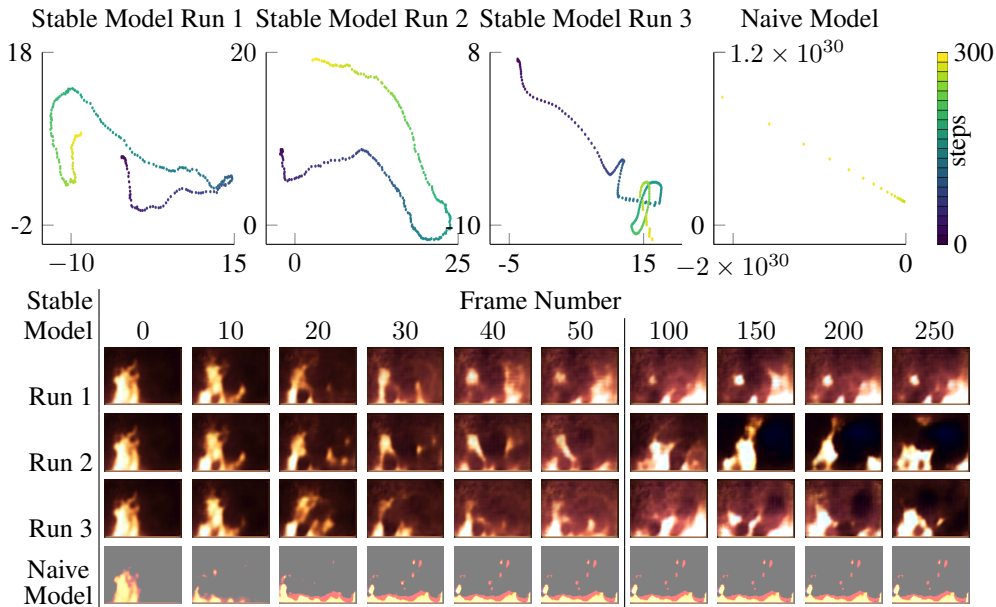

Figure 7: Samples generated by our stable video texture networks, with associated trajectories above. The true latent space is 320-dimensional; we project the trajectories onto a two-dimensional plane for display. For comparison, we present the video texture generated using an unconstrained neural network in place of our stable dynamics model.

model, the dynamics quickly diverge and produce a static image, whereas for our approach, we are able to generate different (stable) trajectories that keep generating realistic images over long time horizons.

## 5 Conclusion

In this paper we proposed a method for learning stable non-linear dynamical systems defined by neural network architectures. The approach jointly learns a convex positive definite Lyapunov function along with dynamics constrained to be stable according to these dynamics everywhere in the state space. We show that these models can be integrated into other deep architectures such as VAEs, and learn complex latent space dynamics is a fully end-to-end manner. Although we have focused here on the autonomous (i.e., uncontrolled) setting, the method opens several directions for future work, such as integration into dynamical systems for control or reinforcement learning settings. Have stable systems as a "primitive" can be useful in a large number of contexts, and combining these stable systems with the representational power of deep networks offers a powerful tool in modeling and controlling dynamical systems.

## Footnotes

[1]Note that the typical softplus smoothed approximation of the ReLU will not work for all purposes above, since we require an activation with $\sigma(0) = 0$

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
