[Reviews · NeurIPS 2019]

Reviewer 1



The authors present a highly original approach for using layered neural networks to learn the function of a continuous time dynamical system while ensuring that the system is stable. The authors leverage recently introduced "input convex neural networks" (ICNNs) and automatic differentiation to simultaneously learn a Lyapunov function to ensure stability as well as train a neural network to approximate a dynamics function. The work is a creative use of ICNNs and automatic differentiation applied to a broad, general problem and one of great significance (learning stable dynamical systems). The approach appears to be a significant improvement (although see below) over previous approaches (which only ensure stability over the data training set). The method itself is clearly presented, with all of the necessary background provided to clearly understand the method, and all of the presented technical material appears correct. The principle weakness of the paper is the clarity and the completeness of the empirical results. The originality of the approach compensates for this weakness, however, the paper would have undoubtedly been scored even more favorably if these results have been more clear and complete. See below for critiques of the clarity fo the empirical results.

Reviewer 2



Originality: The paper describes a formalism to learn lyapunov stable dynamics functions. this is, as acknowledged by the authors, not the first attempt to solve such a problem. However previous methods have either focussed on learning dynamics models linear in embedding space and explicit stability guarantees as a loss. Alternating optimization based methods which use projection constraints have also been used. However this paper proposes a method to learn functions by restricting the function class to lyapunov stable functions with ICNNs -- which is indeed novel and exciting. Overall this would qualify for a novel contribution, of it can be supported amply with experimental evaluation. Quality: The paper presents the methods, formalism and analysis in a convincing manner with sufficient detail. The topic and the contribution is both interesting and non-trivial, even in hindsight. However the results in the experiments section are left a bit wanting. The current results in the paper involve only toy domains, which are not sufficient to convince a reader -- either an expert or a practiioner to adopt this method. Clarity: The paper is written clearly, jumps to the problem statement without much ado and does a good job explaining the background and challenges in the dyanmics learning with stability guarantees. Significance: The results are interesting in particular because of their theoretical value and potential applicability. However the experimental validation is rather insufficient with no baselines.

Reviewer 3



The paper presents a method for constructing neural network architectures that have build-in theoretical guarantees of Lyapunov stability - meaning that the equilibrium will be in the origin and for any initial condition, the network will produce trajectories that converge to the equilibrium. The method is evaluated on the N-link pendulum and video generation problems. The method’s significance comes from two different reasons. First, Lyapunov stability for the system is very difficult to prove with classical methods. Second, deep learning methods are largely empirical, without theoretical guarantees, limiting their applicability for life-critical system. This paper presents a method for learning autonomous dynamics that is guaranteed to be Lyapunov stable, without having the classical toolset. This methodology is original and potentially very useful for many applications, beyond classic controls and videos. For example, protein folding, robotics, weather predictions, material design, etc. The quality of the paper overall is good, although it varies. The theoretical potion is solid. The contribution is clear, well-motivated, and structured well. The empirical validation is somewhat lacking in quality. While the authors are commended for exploring two very different domains (classic controls and video generation), the empirical validation is missing some key elements. For example: - It is not clear how the method would perform on a system without equilibrium, or for that matter the link in the upright initial position. - How do the learned and ground truth models perform in the presence of noise? - Details about the training are missing: 1. methodology for gathering the training set; 2. why the convex network has 60 layers (and in the previous example, it contranied 100 neurons per layer); 3. system info is missing in both examples (equations for the pendulum with the damping factor and reference to the video dataset for the video prediction dataset). 4. Figure 5: Over how many initial conditions was the Figure compiled? Please show error bars. In addition, that authors should include and discuss the following related work: Learning Stabilizable Dynamical Systems via Control Contraction Metrics, Singh et al. WAFR 2018 Continuous Action Reinforcement Learning for Control-Affine Systems with Unknown Dynamics, Faust et al, Acta Automatica Sinica, 2014 The presentation of the paper is excellent. The authors make a theoretical paper very easy to read, and logically introduce one step at the time new notation and the elements of the method. Some minor comments: - Lines 59-60: stating that those conditions are sufficient but not necessary is more clear. - Line 113: What differentiable tools. Please cite? - Line 132-133 - Please, either prove or remove the claims. - Line 145: is V is -> if V is - Line 167: The fact the V -> The fact that V - Line 192: angular velocity \theta -> angular velocity \dot \theta - Please go through the math and use standard notations for vectors (vs scalars). Overall, strong potential theoretical result, with lacking supporting evidence as how well it really works in practice. -------------------------------------------------- Update after author response: Thank you for the response and clarifying the points. The paper presents a strong theoretical contribution, and I leave the score unchanged.

Reviewer 4



Originality: While learning stable dynamical systems has been studied in previous literature in the context of neural networks, the training of an additional Lyapunov function to guarantee stability of an architecture is a novel contribution. Some recent related work which could be included is that of Neural ODEs (Chen et. al, NeurIPS 2018) and stability with recurrent neural networks - i.e. Stable Recurrent Models (Miller & Hardt, ICLR 2019). Quality: The claims are valid and well-supported by theoretical results and empirical analysis. The authors show that their method provably leads to models that are constrained to be stable. Perhaps one area which could be more thorough is that the empirical evaluation of the method is lacking in the form of baselines. The authors only evaluate against a naive neural network. There are potentially several more intelligent baselines, such as penalizing the Jacobian of the network, or even simply clipping the weights. The authors also refer to prior work on penalized losses for training stable networks, but do not evaluate any of the aforementioned methods. Clarity: The paper is well-written and well-organized. Significance: As mentioned by the authors, learning good, stable dynamics model architectures has many downstream applications in reinforcement learning and sequence modeling tasks.

[Author Response · NeurIPS 2019]

Thanks to all the reviewers for their thorough reviews and helpful comments. We appreciate that all the reviewers, even the one most critical of the paper, felt that the fundamental method we introduce here (that of constructing explicitly stable dynamical models, by construction, via the joint dynamics and Lyapunov function), is a novel and exciting methodology.

We also appreciate that the reviewers, especially Reviewer 2, had some concerns with aspects of the papers as well. The main concerns raised were that 1) the method needed further empirical comparisons and 2) the domains studied were rather small/toy domains. To each of these points, we'd like to make the following comments:

**Empirical comparisons.** Reviewer 2 is correct that in all cases the main comparisons here were against a neural network (of the same approximate architecture) with the explicit stability removed. We should point out that, although this reviewer asks us to compare to an RNN-based approach, because we are applying the dynamics recursively, the comparisons we make in all cases (the "simple" or "naive" models in the experimental results) *are* effectively RNNs, just with a structure to make them as comparable as possible to our stable models. If the objection is that the typical RNN has a squashing non-linearity that prevents divergence to infinity, we note that this doesn't actually prevent divergence to extreme values allowed by the nonlinearity (this is very common; we have a squashing function on the output of our VAE). We're happy to include the traditional (e.g., Tanh or LSTM) RNN for comparison, and will add this to the paper; they do not differ substantially from the simple models. We didn't include Embed2Control-style comparisons, as the latent linear modeling there seems largely orthogonal to our main points here, and because we are considering autonomous rather than controlled systems in the current work.

**Toy problems.** We also acknowledge the problems used here are largely meant as demonstrations and thus we felt that the problems, while toy tasks, are still sufficient to illustrate the method. Reviewer 2 felt that the video texture generation was not a good candidate, as a random walk in latent space would also create "videos"; but random walks in latent space result in trajectories that look nothing like the actual dynamics of the system, whereas our modeled trajectories do. We can include a discussion and illustration of this in the revision. And while still a "toy" problem, we felt the 320-dimensional state space of these systems did illustrate the scalability of the approach. However, including an additional spring-damper system is a great idea, and we'll add this to the final paper.

While we certainly agree that these elements can be strengthened, we feel that as demonstrations they highlight the value of the approach, and the overall methodological contribution here is substantial enough to warrant acceptance, as several of the reviewers felt. In addition to these main points, we address a few other comments, and will edit the paper to clarify all these points.

**Reviewer 1** The "simple" model always refers to a simple feedforward network (with the same structure as $\hat{f}$ but without the stability constraints). For the pendulum this is a $2n$-120-120-$2n$ network where $n$ is the number of links in the pendulum.

We'll fully describe the video texture setup in the text (e.g., the source videos are actual videos of physical fire from YouTube) Naive model (again, just same network structure without stability constraints) means that the predicted latent variable diverged to infinity. Thanks for pointing out the confusion here, we'll clarify all of these.

**Reviewer 2** Regarding regularization in general, we can quite easily show that just regularizing the weights of the network is insufficient to achieve stability in practice, unless the system is *extremely* regularized. Additionally, even with regularization this stability is hard to show formally (except locally), and introduces the additional regularization term. However, we'll certainly discuss this point more.

**Reviewer 3** We'll include all these details for the experiments (lack of space to desribe them all here). Thanks very much for pointing on these confusing points.

Thanks also for highlighting these two related works. After going through them a bit, they do seem a bit different in focus (and of course quite different in methodology) as both are concerned with controller design rather than modeling autonomous sytems; but we'll absolutely include and discuss them.

**Reviewer 4** Thanks for highlighting the connections to e.g., recent work on Neural ODEs and work analyzing the stability of RNNs. While these points are somewhat orthogonal, we believe they may actually highlight some nice additional applications of the method (the aforementioned approaches indeed often rely on stability, and thus could be improved by including systems stabilized via our approach).

[Meta-Review · NeurIPS 2019]

The paper has a very strong theoretical contribution which all reviewers appreciated. The opinion about whether the experimental evaluation is sufficient was rather divided. I'll follow the opinion of the majority of the reviewers that in-depth empirical evaluations can (and should) be done in a follow-up paper.